# Alleviate the Drought Stress on *Triticum aestivum* L. Using the Algal Extracts of *Sargassum latifolium* and *Corallina elongate* Versus the Commercial Algal Products

**DOI:** 10.3390/life12111757

**Published:** 2022-11-01

**Authors:** Khadiga Alharbi, Mohamed A. Amin, Mohamed A. Ismail, Mariam T. S. Ibrahim, Saad El-Din Hassan, Amr Fouda, Ahmed M. Eid, Hanan A. Said

**Affiliations:** 1Department of Biology, College of Science, Princess Nourah bint Abdulrahman University, P.O. Box 84428, Riyadh 11671, Saudi Arabia; 2Botany and Microbiology Department, Faculty of Science, Al-Azhar University, Nasr City 11884, Egypt; 3Department of Biochemistry, Faculty of Agriculture, Ain Shams University, Cairo 11566, Egypt; 4Botany Department, Faculty of Science, Fayoum University, Fayoum 63514, Egypt

**Keywords:** water deficit, wheat plant, abiotic stress, algal extract, phytohormone

## Abstract

**Simple Summary:**

Water deficit is the main challenge that has a negative impact on plant physiology and yield production. Moreover, the increased use of chemical fertilizers has a negative impact on the groundwater and ecosystem. The current study investigates the efficacy of two algal extracts versus two commercial algal products to decrease the effect of drought stress on wheat plants under field experiment conditions. The morphological characteristics (shoot length, shoot fresh and dry weight), metabolite contents (chlorophyll, carotenoids), carbohydrate contents, protein contents, and antioxidant enzymes (peroxidase, superoxide dismutase, and polyphenol oxidase) were estimated after two stages and compared with unstressed plants. Moreover, acidic phytohormones (IAA, GA3, and ABA) and yield characteristics were detected.

**Abstract:**

Herein, two seaweed extracts (*Sargassum latifolium* and *Corallina elongate)*, and two commercial seaweed products (Canada power and Oligo-X) with a concentration of 5% were used to alleviate the drought stress on wheat plants. The extract of *C. elongate* had the highest capacity to ameliorate the deleterious effects of water scarcity followed by *S. latifolium* and the commercial products. The drought stress reduced wheat shoots length and the contents of pigments (chlorophyll and carotenoids), carbohydrates, and proteins. While the highest increment in the total carbohydrates and protein contents of the wheat shoot after two stages, 37-and 67-days-old, were noted in drought-stressed plants treated with *C. elongate* extract with values of (34.6% and 22.8%) and (51.9% and 39.5%), respectively, compared to unstressed plants. Decreasing the activity of antioxidant enzymes, peroxidase, superoxidase dismutase, and polyphenol oxidase in drought-stressed plants treated with algal extracts indicated amelioration of the response actions. Analysis of phytohormones in wheat plants exhibited increasing GA3 and IAA contents with percentages of (20.3–13.8%) and (72.7–25%), respectively. Interestingly, all morphological and metabolic characteristics of yield were improved due to the algal treatments compared with untreated drought-stressed plants. Overall, the algal extracts, especially those from seaweed of *C. elongate,* could represent a sustainable candidate to overcome the damage effects of water deficiency in the wheat plant.

## 1. Introduction

Drought is considered the main restricting crop production due to its serious effects on plant growth, plant respiration, stomatal movement, and photosynthesis, leading to disruption of plant metabolism and physiology [1,2]. Under drought stress, various defense mechanisms are activated, such as structural and morphological changes, production of phytohormones, resistant gene expression, and the production of active substances with osmotic regulatory properties [3]. Some water deficit signals, such as Ca^2+^, abscisic acid, adenosine-5-diphosphate-ribose, and inositol-triphosphate, are synthesized, leading to changing the physiological and morphological plant traits via signal transduction [4]. Drought stress signals can affect the expression of some functional and downstream gene products, such as soluble sugar, proline, glycine betaine, and aquaporin, which have crucial functions in metabolism, leading to negative impacts on the plant. The production of phytohormones and some signal molecules, such as gibberellic acids, auxins, ethylene, abscisic acid, salicylic acid, cytokinins, jasmonic acid, and nitric oxide, are involved in the resistance of plants to drought stress [5,6].

Wheat (*Triticum aestivum* L.) is considered one of the most important crops all over the world. With year-over-year cultivation particularly in Egypt, the nation’s wheat production has reached over 8 million tons annually [7]. However, the yield from wheat planting does not meet the domestic use. Therefore, it is necessary to search among the high-yield cultivars with cutting-edge irrigation systems and innovative techniques that optimize the use of water and fertilizer. Regarding varieties, numerous researchers have noted that some species and varieties are better adapted to poor soil conditions than others and consequently their nutritional requirements significantly differed from others [7,8]. Agriculture is facing the twin challenges of growing crop production and global climate change. Rising temperature, drought, salinity, floods, geological processes, and weather extremes adversely affect agriculture, particularly in the developing world [9]. The introduction of resistance to salt, cold, and drought into crop plants has thus become a subject of major economic interest for agriculture.

Natural products can be used to increase the tolerance of the plant to various environmental disturbances, algal extracts, and their products. The algal metabolites can increase the seedling’s development and seed germination and improve plant tolerance toward various biotic and abiotic stresses [10,11]. Moreover, the algal extracts have the efficacy to improve the plant growth and yield of several crops, including different grasses, cereals, flowers, and vegetable species [12]. Liquid extracts made from seaweed have become more important as foliar sprays and soil. They are used to promote seedling germination and root growth as well; therefore, using seaweeds as plant bio-stimulants is now one of the most promising biotechnological tools in agronomy. For instance, foliar spray and soil drenching with aqueous extracts of *Sargassum johnstonii* increased the tomato plant height, shoot length, root length, and branch count as well as reproductive parameters (flower number, fruit number, and fresh weight) [13]. Moreover, the crude extract from different seaweeds, viz. *Ulva lactuca, Cladophora dalmatica, Enteromorpha intestinalis, Corallina mediterranea, Pterocladia pinnate*, and *Janiarubens* collected from the Egyptian Mediterranean Sea coast were used as a bio-stimulant for the growth of *Vicia faba* L. [14]. The authors reported that the highest seed germination, root and shoot length, and amounts of lateral roots were attained after the treatment with *C. dalmatica* extract. Additionally, all of the seaweed’s raw extracts boosted the amount of protein, total soluble carbohydrates, and chlorophyll in the leaves as well as in the root and shoot systems as compared with the control.

Thus, the current study aims to evaluate the efficacy of algal extract from two seaweeds (*Sargassum latifolium* and *Corallina elongate*) in comparison with two commercial seaweed products (Canada power and Oligo-X) to alleviate the drought stress on the wheat plant under field experiment conditions by foliar spray method. Various parameters, such as shoot length, fresh and dry weight, pigmentation, carbohydrate and protein contents, antioxidant enzymatic activity, phytohormones, and yield parameters, which indicate the efficacy of algal extract to alleviate the drought stress, were investigated at two stages, after 37 and 67 days of sowing.

## 2. Materials and Methods

### 2.1. Plant Material and Treatments

Seeds of wheat plants (*Triticum aestivum* L.) (Var. suds 1) were obtained from the Agricultural Research Centre, Ministry of Agriculture, Giza, Egypt. Commercial algal products (Canada power and Oligo-X) were purchased from the Egyptian Canadian for humate technology and agriculture consultancy, Egypt. *Sargassum latifolium* (Turner) C. Agardii was collected from the Hurgada Red Sea coast (27°08′99.8″ N 33°86′20.5″ E) and Baltim City (31°56′09.88″ N: 31°31′01.1″ E) during May 2020, whereas the *Corallina elongate* J. Ellis was collected during the time of April 2020 from shallow water beside the shore of the Mediterranean Sea at Abou Quair coast, Alexandria, Egypt (31°32′62.66″ N 30°06′00.42″ E).

### 2.2. Preparation of Algal Extract

For fresh seaweed algae, the collecting mass was washed thrice with tap water to remove any adhering particles followed by oven drying at 60 °C for five hours and hand- pulverizing to form a powder. Approximately, 5 g of the prepared powder was well mixed with 100 mL of distilled H_2_O before being heating at 60 °C for 25 min under stirring conditions. After that, the previous mixture was centrifuged for 15 min at 5000 rpm, and the collected supernatant was kept in the refrigerator at 4 °C till being used [15]. The commercial algal extract (Canada power and Oligo-X) was prepared before being used by adding 5 mL of extract to 100 mL of distilled H_2_O in accordance with the instructions from the manufacturer. The prepared extracts were used as a foliar treatment (spraying method) after 30 and 60 days from sowing.

### 2.3. Treatments and Experimental Design

The field experiment was conducted under field conditions at the Faculty of Science Garden, Al-Azhar University, Cairo, Egypt to investigate the effect of foliar treatment of algal extract on the growth of wheat under drought stress. The soil used in the field experiment was loamy soil containing sand: silt: and clay with percentages of 95.2%, 3.5% and 1.4 %, respectively. Moreover, Ca, Na, K, Cl, and P were present in soil with values of 25, 185.3, 16.2, 132.5, and 24.4 mg Kg^−1^, respectively. The seeds of wheat were subjected to surface sterilization before being sowed by immersion in 70% ethyl alcohol for one minute followed by sodium hypochlorite (4%) for 15 min and washing with sterilized distilled H_2_O [16,17].

Uniform seeds were selected and planted in the soil under field conditions in a randomized block design (RBD). The randomized design was established with five blocks, and within each block, there were six plots for the various treatments as follows: control (irrigated by tap water every 14 days in absence of algal extract), drought stress (irrigated by tap water every 28 days in absence of algal extract), plants growing under drought stress and sprayed with *Corallina* extract, *Sargassum* extract, Canada power, and Oligo-X. The seeds were planted on one side of the ridge and approximately 10 cm between each hill and the other. The irrigation of wheat was achieved accordance to the instructions obtained with seeds from Egyptian Agricultural Ministry. The plants were sprayed twice with the above-mentioned treatments, after 30 and 60 days of plant age. The samples for different analyses were taken after 37 (stage I) and 67 days of sowing (stage II). For analysis, 3–10 plant samples for analysis were randomly collected per treatment [18].

### 2.4. Growth Traits Measurement

#### 2.4.1. Morphological Parameters Measurement

The shoots of the wheat plant after the first stage (37-day-old) and second stage (67-days-old) were collected for the measurement of shoot length (cm) and shoot fresh weight. The collected samples were subjected to oven-drying at 70 °C till constant weights were obtained to measure the shoot dry weight. Water content percentage in the shoots was estimated based on the following equation:(1)$$Water\;content(\%)=\frac{M_{f}-M_{d}}{M_{f}}\times 100$$
where M_f_ is the fresh mass and M_d_ is the dry mass after drying the shoots in an oven.

Fresh shoot samples were stored in a refrigerator for the measurement of enzymatic activities.

#### 2.4.2. Photosynthetic Pigments Measurement

The photosynthetic pigments including chlorophyll a (Chl-a), chlorophyll b (Chl-b), and carotenoids in different treatments were estimated according to Strain and Svec [19]. Briefly, approximately one gram of fresh wheat leaves was cut into fine pieces and mixed with 100 mL of aqueous acetone solution (80%) and grinding together in a mortar containing glass powder. After that, the homogenate solution was filtered using Whatman No. 1 and transferred filtrate to a volumetric flask, adding aqueous acetone solution (80%) to get a final volume of 100 mL. The optical density (OD) of the previous mixture was measured using a JENWAY spectrophotometer at two wavelengths, 649 nm and 665 nm, which characterized the maximum absorption of Chl-a and Chl-b, respectively. The concentration of chlorophyll in wheat plant tissues was calculated by the following equations:(2)Chl-a/g plant tissue=11.63(OD at 665)−2.39(OD at 649)
(3)Chl-b/g plant tissue=20.11(OD at 649)−5.18(OD at 665)
(4)Total Chlorophyll(a+b)/g plant tissue=6.45(OD at 665)+17.72(OD at 649)

The concentration of carotenoids content was calculated according to the following equation [20]:(5)$$\begin{array}{l} Carotenoid\;contents\;(mg/g\;fresh\;weight)=\\ 1000\;X\;(OD\;at\;470)-1.82(Chl-a)-85.02(Chl-b)/198 \end{array}$$

#### 2.4.3. Total Soluble Carbohydrates and Proteins

The carbohydrate content in the dry wheat shoot was assessed according to the method of Umbreit et al. [21], whereas the soluble protein content was estimated based on the method of Lowery et al. [22].

#### 2.4.4. Determination of Antioxidant Enzymes

Approximately 5 g of fresh wheat leaves (first and second leaves and terminal part) were soaked into sodium phosphate buffer (50 mM) at a pH of 7 mixed with polyvinyl pyrrolidine (PVP, 1.0%). The mixture was homogenized and centrifuged for 20 min at 20,000 rpm under cooling conditions (4 °C). the supernatant was collected and used for the determination of enzymatic activity [23]. The activity of peroxidase enzymes (EC 1.11.1.7) was calculated according to Kar and Mishra [24]. In this method, the assay solution contains phosphate buffer (125 µmoles, pH 6.8), pyrogallol (50 µmoles), hydrogen peroxide (H_2_O_2_, 50 µmoles), and diluted enzyme extract (1 mL and diluted up to 20-times). The mixture was incubated at 25 °C for five minutes followed by adding 0.5 mL H_2_SO_4_ (5%, *v*/*v*) to stop the reaction. The color intensity due to the formation of purpurogallin was determined by measuring their absorbance at a wavelength of 420 nm and expressed as the enzyme unit (EU)/mg protein. The activity of polyphenol oxidase (EC 1.10.3.1) was assessed in a solution containing the same components of peroxidase except H_2_O_2_ under the same conditions [24]. The absorbance of formed color due to the formation of purpurogallin was measured at a wavelength of 420 nm and expressed as the EU/mg protein.

The activity of superoxide dismutase (EC 1.15.1.1) was assayed according to the method described by Bayer and Fridovich [25] through the photoreduction of Nitroblue-tetrazolium (NBT). The activity of superoxide dismutase was displayed as EU/mg protein.

#### 2.4.5. Extraction and Determination of Endogenous Acidic Phytohormones

The extraction and quantitative estimation of endogenous acidic phytohormones (Indole-3-acetic acid (IAA), Gibberellic acid (GA3), and Abscisic acid (ABA)) were achieved at the samples collected from the first stage (37-days-old) according to the method described by Yang et al. [26]. In this method, 1 g of terminal plant part was mixed well with acetonitrile (10 mL), liquid nitrogen, and diethyldithiocarbamate (50 mg L^−1^) before being incubated at 4 °C for 12 h in dark conditions. At the end of incubation conditions, the mixture was centrifuged at 10,000 rpm for 15 min. The supernatant was collected and concentrated by a rotary evaporator to form a residue which re-dissolved in 10 mL of phosphate buffer (0.5 mol L^−1^, pH 8) and mixed with 8 mL of chloroform. The previous mixture was shaken well and removed from the chloroform phase followed by adding 0.15 g of insoluble polyvinylpyrrolidone to the aqueous phase before being centrifuged at 10,000 rpm for 15 min. Approximately 5 mL of supernatant (adjusted their pH at 3 by pure formic acid) was mixed with 5 mL of ethyl acetate, shaken well, and discarded in an aqueous layer. The ethyl acetate phase was concentrated by a rotary evaporator and dissolved the residue in 1 mL of acetonitrile: 6% acetic acid: methanol (5:45:50, *v*:*v*:*v*). In the end, the hydrophobic membrane (0.22 mm) was used to filter the last mixture that was used to determine plant hormones using high-performance liquid chromatography (HPLC) [18].

#### 2.4.6. Yield Analysis

The weight of spike (g), spike length (cm), the number of grains per spike, and weight of grains (g) per spike were determined after 165 days of sowing. The total soluble carbohydrate and protein in the grain yield were calculated from seven single plants of the row in each plot.

### 2.5. Statistical Analysis

Results in the current study were statistically analyzed using SPSS v18 (SPSS Inc., Chicago, IL, USA). The analysis of variance (ANOVA) test was used for multiple sample comparison followed by a multiple comparison Tukey’s test.

## 3. Results and Discussion

### 3.1. Effect of the Foliar Spray of Aqueous Algal Extracts on the Growth of Wheat Plants under Drought Stress

Data recorded in Figure 1 and Table 1 show the impact of foliar spraying of algal extracts represented by two commercial products versus two fresh seaweed extracts of on the growth traits of common *Triticum aestivum* L. under draught stress. The treatment with foliar spraying has advantages over other methods due to the symmetrical spreading of the extract over the whole plant. Moreover, the response of plant to various treatments was enhanced with foliar spraying [27]. Sarker and co-authors reported that the losses of micronutrients that were added to plants via foliar spraying were decreased as compared with those lost by other treatment methods [28]. Water is essential for the growth of plants. Contrarily, drought refers to the lack of sufficient moisture for a plant to grow normally and finish its life cycle [29]. Restricted nutrient and water acquisition are among the many effects of drought on plant development that are widely acknowledged [30].

According to the obtained results, there is a significant decrease in the lengths, fresh, and dry weights of the plant shoot in water stress compared to the unstressed control. Also, the results revealed that the seaweed extracts and commercial algae under stress conditions enhanced these parameters in most cases, at the two stages after 37 and 67 days. The maximum enhancement in the morphological parameters (lengths, fresh, and dry weight of shoot) was observed in the presence of *C. elongate* algae extracts. As shown, the shoot length under draught stress and treated with *C. elongate* extract was 49.8 ± 0.8 cm and 76.7 ± 1.5 cm after the first and second stage, respectively, as compared with control (untreated drought-stressed plants) which was 41.7 ± 1.2 cm and 61.2 ± 0.9 cm after the same stages. Interestingly, there is no significant difference between the shoot length under normal growth (51.7 ± 1.9 (first stage) and 77.2 ± 1.2 cm (second stage) and the length of the plant grown under drought stress and sprayed with *C. elongate* extract (Figure 1). On the other hand, similar shoot lengths were obtained for plants sprayed with other algal extracts after the second stage (Figure 1).

Data in Table 1 show that the fresh and dry weights of wheat shoots after the first and second stages were (5.5 ± 0.7 and 1.6 ± 0.02 g) and (9.1 ± 0.8 and 2.1 ± 0.3 g) for unstressed control plants. These values were decreased under drought stress to (4.04 ± 0.4 and 0.9 ± 0.001 g) and (7.9 ± 0.4 and 1.8 ± 0.2 g) after the first and second stage, respectively. The highest water shoot content of 78.2% was estimated in plants treated by Canada powder. Interestingly, the negative impacts of drought stress on fresh and dry weights were highly decreased in the presence of the algal aqueous extract of *C. elongate*, followed by *S. latifolium*, Canada powder, and Oligo-X, respectively (Table 1).

According to this theory, the addition of blue-green algae enhances the flag leaf area and increases the plant length of rice (*Oryza sativa*) [31]. The improvement in glycine betaine concentration in treated plants can be linked to the significance of seaweed extract in the stress water effect. Glycine betaine, alanine betaine, and proline betaine have been found to positively correlate with leaf osmotic potential in numerous plant species [32]. It is now understood that these chemical substances also have cellular osmo-protective properties [33]. Seaweed solution prepared from *Ecklonia maxima* (Kelpak) was found to increase the shoot growth in three species of *Eucalyptus* [34]. The efficacy of algal extract to ameliorate the negative impacts in plants under stress could be attributed to the relatively high concentrations of indoles present in the extract [35]. The increases in the growth parameters due to algal extract treatment also coincide with the results of Kumari et al. [13], who demonstrated that the vegetative growth (plant height, shoot length, root length, and the number of branches) and reproductive parameters (flower number, fruit number, and fresh weight) of tomato plant were improved by aqueous extracts of *Sargassum johnstonii* at concentrations from 0.1 to 0.8% (*w*/*v*), which is equivalent to 1–8 mg mL^−1^. In the current study, the enhancement of growth parameters of wheat plants by the spraying of commercial or fresh algal extracts may be due to the presence of cytokinin’s, minerals, and various nutrients in the prepared extracts. These ingredients are known to improve the growth and increase cell division and cell enlargement [36,37]. Compatible with the obtained results, Mansori et al. [38] showed that the growth and yield of *Cajanus cajan* (L.) Millsp., *Vigna sinensis* L., and *Phaseolus vulgaris* L. cultivated under water stress were improved by the foliar spray of an algal extract. Briceo-Domnguez et al. [39] reported that the aqueous extract of brown algae, *Macrocystis pyrifera* has the plant growth promoting activity for improve the growth of tomato (*Solanum lycopersicum)* under various abiotic stresses.

### 3.2. Effect of Algal Extracts on the Pigments Contents of Wheat Plants Grown under Drought Stress

Results recorded in Table 2 reveal that Chl-a, Chl-b, Chl-(a + b) and carotenoid contents of wheat leaves decreased as a result of water deficit. Herein, the pigment contents including Chl-a, Chl-b, and total Chl. (a + b), and carotenoids for wheat grown under drought stress were decreased with percentages of (54.4%, 20.8%, 39.7%, and 14.3%) and (23%, 17.5%, 20%, 15.4%) for first and second stages, respectively as compared with unstressed control plants. This reduction may be due to the directing of most of the plant’s energy to produce water deficit-resistant materials, such as proline and phenols [40]. All treatments appeared enhancement of pigment contents, especially *C. elongate* extract treatment which showed the highest improvement in these parameters. The foliar treatment of wheat grown under drought stress by *C. elongate* algal extract increased the chlorophyll a + b by 34.8% and 22.8% at first and second stages, respectively, as compared with unstressed plants. Moreover, the total chlorophyll a + b increased by the percentage of 28.6% at the second stage due to foliar spraying with commercial algal extract Olig-X compared with untreated drought-stressed plants (Table 2). Overall, plants grown under stress minimized the chlorophyll synthesis rate as reported previously [41]. In a similar study, the chlorophyll content of brinjal (*Solanum melongena*) leaves was increased by a percentage of 79% when brown algal extracts were applied at a spray rate of 1.5% compared to plants that were unsprayed [42]. The beneficial effect of algal extracts in protecting chlorophyll degradation could be attributed to betaine and betaine-like compounds present in seaweed [43]. Numerous beneficial components, including vitamins, minerals, amino acids, and indole-acetic acid, are present in the algal extract and may help with the synthesis of photosynthetic pigments [44].

### 3.3. Effect of Foliar Spraying of the Algal Extract on the Carbohydrates and Protein Contents

Drought stress caused a significant decrease in the total soluble carbohydrates and protein contents at the first and second stage as shown in Table 3. Data analysis showed that the foliar spraying of wheat grown under drought stress with algal extract of *C. elongate* increased the carbohydrate and protein contents of shoots with percentages of (34.6% and 22.8%) and (51.9% and 39.5%) after the first and second stages, respectively as compared with a plant grown under drought stress without any treatment. Interestingly, the foliar spraying of various algal extracts increases the carbohydrate and protein contents of the shoot compared to a wheat plant grown under normal conditions (control) (Table 3). For instance, the spraying of a commercial product of algal extract, Olig-X, increases the carbohydrate content of the shoot with percentages of 11.3% and 3.3% for the first and second stage respectively as compared with normal control and with percentages of 30.7% and 19% compared with a plant grown under drought stress (Table 3). The obtained results corroborate with those reported by Ali et al. [45], who reported that kernel sugar and protein contents were reduced in water-stressed maize plants. In contrast, the total carbohydrates and total soluble protein in shoots increased under drought conditions as previously reported [46,47]. In general, the effect of algal extracts has a vital effect in decreasing the adverse effect of drought on plant chemical constituents. In addition, algal extract is rich in major and minor nutrients, amino acids, and phytohormones which are important in reducing the harmful effects of drought [48,49]. In a similar study, seaweed extract at 4 mL L^−1^ demonstrated the efficacy of increasing the total carbohydrate and crude protein concentrations in leaves as well as in seeds of common bean [50].

### 3.4. Antioxidant Enzymatic Activity of Wheat Plant Grown under Drought Stress

The demonstrated results in Table 4 showed that the antioxidant enzyme activity of wheat plants at both stages increased in response to water stress. However, all treatments, especially *C. elongate* extract treatment, showed a significant decrease in these activities. Our result may be explained by the effect of seaweed extract in reducing cell damage caused by ROS as reported previously [43]. On the other hand, the application of seaweed extract to turf grasses increased the activity of the antioxidant enzyme superoxide dismutase (SOD), which scavenges superoxide [51]. Many researchers have reported that seaweed extracts enhance the ascorbate peroxidase activities [52], demonstrating the strong antioxidant properties of seaweeds which have been correlated to bioactive compounds [53]. Polyphenols represent a large family of plant secondary metabolites. The synthesis of these compounds is induced in response to biotic and abiotic stimuli, such as drought, chilling, ozone, heavy metals, attack by pathogens, wounding, and nutrient deficiency [54]. Polyphenols may act as antioxidants to protect the plant against oxidative stress [55]. Water stress is associated with increased oxidative stress caused by a buildup of reactive oxygen species (ROS), primarily O_2_^−^ and H_2_O_2_, in mitochondria, chloroplasts, and peroxisomes. The activation of antioxidant enzymatic activity is a common strategy that plants use to reduce oxidative stress [56,57].

### 3.5. Acidic Growth Hormones

In the current study, acidic growth hormones (IAA, GA3, and ABA) exhibited increases in GA3 and IAA contents of wheat plants as a result of all treatments in comparison to a stress condition, whereas the ABA contents decreased. The highest values of IAA and GA3 under stress were obtained with *C. elongate* treatment with values of 1.1 ± 0.1 and 9.4 ± 0.4 mg/100 g, respectively as compared with stress conditions (0.3 ± 0.05 and 7.5 ± 0.5 mg/100 g). On the other hand, The ABA concentration was decreased from 0.9 ± 0.01 µg/100 g to 0.4 ± 0.03, 0.6 ± 0.05, 0.3 ± 0.08, and 0.3 ± 0.09 µg/100 g as a result for spraying algal extract of *C. elongate*, *S. latifolium*, Canada power, and Oligo-X, respectively (Table 5). The obtained data, especially for those of ABA confirmed that the foliar spraying of various algal extracts alleviated the stress caused by water deficit. Phytohormones play vital roles in the drought tolerance of plants. Auxins induce new root formation by breaking root apical dominance induced by cytokinin. As a prolific root system is vital for drought tolerance, auxins have an indirect but key role in this regard. IAA (indole-3-acetic acid) is one of the most multi-functional phytohormones and is vital not only for plant growth and development, but also for governing and coordinating plant growth under stress conditions [58,59].

The clearance of reactive species and pathway of signals transduction is considered the main mechanisms to ameliorate drought stress response through the secretion of phytohormones such as jasmonic acid, abscisic acid, auxin (IAA), salicylic acid, gibberellin, ethylene, and brassinosteroids, in addition to calcium [60]. The role of IAA in ameliorating drought stress response can be attributed to the overexpression of late-embryogenesis abundant genes that are responsible for resistance enhancing of the plant to water deficit. Also, auxin/IAA increases the resistance of plants toward drought stress via interacting with other plant hormones. For instance, the auxin regulates the expression of the aminocyclopropane-1-carboxylate synthase gene which is responsible for the biosynthesis of ethylene. This interaction promotes the resistance of plants to drought stress [61]. In plant cells, seed germination, cell division, and cell and stem elongation are mainly regulated by the production of gibberellic acids. Under drought stress, decreasing the content of GA leads to the accumulation of specific proteins called DELLA, which is identified as repressors for seed germination, plant growth, cell development, and flowering [62]. In the current study, the application of algal extracts has the efficacy to return GA to the normal value as possible compared to a wheat plant grown under normal conditions (Table 4). The addition of GA3 to maize plants was used to alleviate the water stress by protecting the membrane permeability, improving the content of pigments, and keeping the leaf water content [63]. The treatment of wheat plants with a low dose of GA3 enhanced seed germination under water deficit [64]. ABA is one of the plant hormones that provide a high resistance toward various abiotic stresses, such as drought, cold, temperature, and salt [65]. ABA is accumulated in plant cells under stress and acts as a trigger to overcome the negative impacts due to unfavorable conditions. Under drought stress, the ABA is formed in plant roots and transferred to leaves to increase the resistance of the plant against water scarcity through stomata closure and growth reduction [66]. Moreover, ABA under drought stress can enhance the expression of various stress-responsive genes that protect plants [67].

### 3.6. Yield Parameter Measurement

Our results showed that the fresh and commercial algal extracts caused remarkable improvement in the yield parameters compared to drought stress (Table 6). The highest values were recorded of wheat plants grown under drought stress and treated with algal extract of *C. elongate* followed by *S. latifolium*, Canada powder, and Oligo-X. These results are in agreement with those obtained by Hernández-Herrera et al. [68], who reported that the use of biostimulants, often based on natural extracts such as seaweeds, has been proposed as a sustainable strategy for improving crop yields without adversely impacting the environment. Algal extract from *C. elongate* seaweeds increases the weight of spike (g), spike length (cm), the number of grains/spike, and weight of grains (g)/spike by percentages of 50%, 23.3%, 29.1%, and 50%, respectively as compared with wheat grown under drought stress without any treatment. In a recent study, the application of 9% of algal extract to onion grown under water stress increased the bulb yield by a percentage of 67% and 102%, in the first and second seasons [69]. The production of onions was dramatically reduced when irrigation water was cut back. However, the application of an algal extract improved onion development and lessened the impacts of water stress. Moreover, the spraying of algal extract of *Ascophyllum nodosum* enhanced the growth of soybean and lessened the impact of drought effects [70].

In addition, foliar spraying with algal extract ameliorates the negative effects of drought stress and improves the total carbohydrates and crude protein in the yield. The highest increment in total carbohydrates and proteins in the seeds was noted in drought-stressed plants treated with *C. elongate* with percentages of 27.9% and 40.6%, respectively, compared with a plant grown in drought stress in absence of treatments. Interestingly, the spray of wheat plants with algal extract of *C. elongate* increases the carbohydrate and protein in drought-stressed plants with percentages of 3% and 21.9% compared with control (wheat grown under normal conditions) (Figure 2). Overall, the presence of various algal extracts has a positive impact on the carbohydrate and protein content of seeds.

The decrease of the total soluble carbohydrates and proteins of seeds under drought stress may be related to the reduction of enzymes and metabolic activities which reduces the movement of different nutrients to seeds. Similarly, the carbohydrates and proteins were significantly decreased in seeds of drought-stressed quinoa plants [71]. Similar suggestions were reported by Agami [72], who found that the total soluble proteins contents on lettuce were significantly decreased in drought-stressed (60% FC) plants compared with non-water-stressed plants. Further, El-Saadony et al. [73] reported that the protein and total carbohydrate of seeds of pea plants grown in sandy soil were significantly decreased under water-deficit conditions (30% FC). Overall, the treatment of wheat plants with extract of fresh seaweed especially *C. elongate* was more suitable to ameliorate the drought stress than commercial products. This phenomenon could be attributed to the huge metabolites secreted by *C. elongate,* such as soluble carbohydrates, polysaccharides, lipids, and amino acids, which have a promising role as osmoregulatory agents [74,75].

## 4. Conclusions

In light of the present study, spraying wheat plants with *Carolina* algal extract can successfully ameliorate the deleterious effects of drought stress as well as enhance the plant growth performance and yield parameters. Furthermore, it is worth noting that *Sargassum* and *Carolina* extracts were more effective than commercial algal treatments in raising the plants’ tolerance to drought. Therefore, we would venture to recommend the use of spraying wheat plants with *Carolina* extract as a new, natural, and low-cost method for not only the alleviation of drought stress on plants, but also for stimulating growth with no discernible adverse effects.

## Figures and Tables

**Figure 1 life-12-01757-f001:**
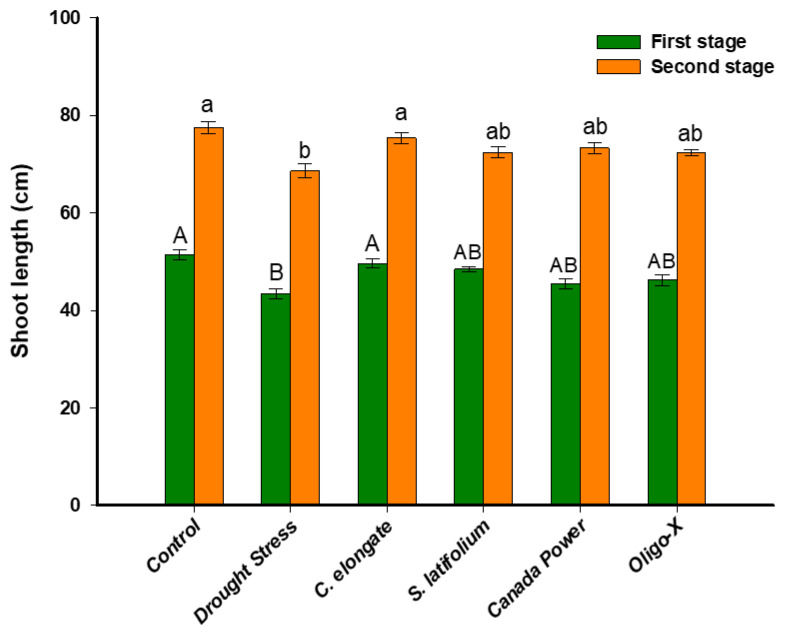
Shoot lengths of wheat plants grown under drought stress and foliar spraying with algal aqueous extracts as compared with control (growing under normal growth and irrigated every 14 days). Error bars are means ± SE. Different letters on bars denote that mean values are significantly different (*p* ≤ 0.05) by Tukey’s test.

**Figure 2 life-12-01757-f002:**
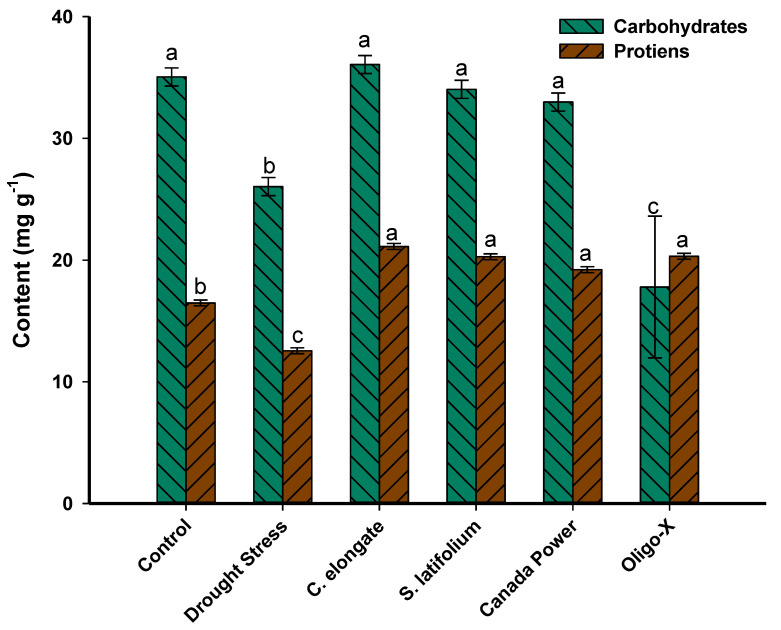
The total carbohydrate and protein content in seeds of wheat grown under drought stress and sprayed with various algal extracts. Error bars are means ± SE. Different letters on bars denote that mean values are significantly different (*p* ≤ 0.05) by Tukey’s test.

**Table 1 life-12-01757-t001:** Fresh and dry weights and water shoot content of wheat plant growing under drought stress and spraying with aqueous algal extract after two stages (first stage at 37-days and second stage at 67-days of sowing). Different letters between columns denote that mean values are significantly different (*p* ≤ 0.05) by Tukey’s test, values are means ± SE.).

Treatments	Shoot Fresh Weight (g)		Shoot Dry Weight (g)		Water Shoot Content (%)	
	First Stage	Second Stage	First Stage	Second Stage	First Stage	Second Stage
Control	5.5 ± 0.7 ^a^	9.1 ± 0.8 ^a^	1.6 ± 0.02 ^a^	2.1 ± 0.3 ^a^	71.1 ± 0.7 ^c^	76.5 ± 0.8 ^a^
Drought Stress	4.04 ± 0.4 ^b^	7.9 ± 0.4 ^b^	0.90 ± 0.001 ^b^	1.8 ± 0.2 ^a^	75.6 ± 0.2 ^b^	77.2 ± 0.5 ^a^
*C. elongate*	5.7 ± 0.2 ^a^	9.1 ± 0.7 ^a^	1.6 ± 0.1 ^a^	2.1 ± 0.2 ^a^	71.5 ± 0.4 ^c^	77.2 ± 0.4 ^a^
*S. latifolium*	4.7 ± 0.2 ^ab^	8.2 ± 0.4 ^b^	1.5 ± 0.02 ^a^	1.9 ± 0.2 ^a^	66.4 ± 1.8 ^d^	76.4 ± 0.5 ^a^
Canada Power	4.9 ± 0.3 ^ab^	7.9 ± 0.4 ^b^	1.1 ± 0.05 ^ab^	1.9 ± 0.2 ^a^	78.2 ± 1.4 ^a^	76.6 ± 0.9 ^a^
Oligo-X	4.2 ± 0.2 ^b^	8.04 ± 0.3 ^b^	1.1 ± 0.1 ^ab^	1.9 ± 0.1 ^a^	72.9 ± 0.6 ^c^	76.1 ± 0.1 ^a^

**Table 2 life-12-01757-t002:** Chlorophyll content of wheat leaves (mg g^−1^ fresh weight) as affected by algal extract treatments under normal and stress irrigation. Different letters between columns denote that mean values are significantly different (*p* ≤ 0.05) by Tukey’s test, values are means ± SE.

Treatments	Chlorophyll (a)		Chlorophyll (b)		Chlorophyll (a + b)		Carotenoids	
	First Stage	Second Stage	First Stage	Second Stage	First Stage	Second Stage	First Stage	Second Stage
Control	6.8 ± 1.9 ^a^	7.4 ± 0.6 ^a^	5.3 ± 1.8 ^a^	9.1 ± 1.6 ^a^	12.1 ± 3.7 ^a^	16.5 ± 1.1 ^a^	2.8 ± 0.8 ^a^	1.3 ± 0.33 ^b^
Drought Stress	3.1 ± 1.5 ^b^	5.7 ± 0.1 ^c^	4.2 ± 2.9 ^b^	7.5 ± 0.5 ^b^	7.3 ± 1.4 ^c^	13.2± 0.4 ^b^	2.4 ± 0.5 ^a^	1.1 ± 0.13 ^b^
*C.* *elongate*	5.9 ± 1.8 ^a^	7.5 ± 0.5 ^a^	5.3 ± 2.7 ^a^	9.5 ± 1.6 ^a^	11.2 ± 1.7 ^a^	17.1 ± 1.04 ^a^	2.1 ± 1.6 ^a^	1.01 ± 0.3 ^b^
*S. latifolium*	4.2 ± 0.1 ^ab^	6.7 ± 0.02 ^b^	4.9 ± 0.1 ^ab^	9.9 ± 0.75 ^a^	9.1 ± 0.04 ^b^	16.7 ± 0.8 ^a^	1.0 ± 0.6 ^b^	1.2 ± 0.5 ^a^
Canada Power	3.8 ± 2.6 ^b^	6.6 ± 0.02 ^b^	4.3 ± 3.3 ^b^	10.2 ± 0.1 ^a^	8.1 ± 5.9 ^c^	16.8 ± 0.2 ^a^	1.5 ± 0.3 ^ab^	1.5 ± 0.2 ^ab^
Oligo-X	3.4 ± 1.4 ^b^	6.6 ± 0.03 ^b^	4.4 ± 1.01 ^b^	11.9 ± 1.1 ^a^	7.8 ± 2.5 ^c^	18.5 ± 1.1 ^a^	0.8 ± 0.5 ^b^	1.9± 0.7 ^a^

**Table 3 life-12-01757-t003:** Effects of drought stress, fresh algal extracts, and commercial algae products on the carbohydrate and protein contents in the shoot of wheat. Different letters between columns denote that mean values are significantly different (*p* ≤ 0.05) by Tukey’s test, values are means ± SE.

Treatments	Carbohydrate Shoot (mg g^−1^ DW)		Protein Shoot (mg g^−1^ DW)	
	First Stage	Second Stage	First Stage	Second Stage
Control	13.3 ± 0.4 ^ab^	11.7 ± 0.4 ^a^	4.9 ± 0.2 ^b^	6.4 ± 0.1 ^a^
Drought Stress	10.4 ± 0.4 ^b^	9.8 ± 0.1 ^a^	3.8 ± 0.1 ^c^	4.6 ± 0.1 ^b^
*C. elongate*	15.9 ± 0.1 ^a^	12.7 ± 0.1 ^a^	7.9 ± 0.3 ^a^	7.6 ± 0.3 ^a^
*S. latifolium*	12.9 ± 0.1 ^b^	12.5 ± 0.1 ^a^	6.7 ± 0.1 ^a^	6.9 ± 0.1 ^a^
Canada Power	14.0 ± 0.2 ^a^	12.5 ± 0.2 ^a^	5.6 ± 0.2 ^b^	6.7 ± 0.1 ^a^
Oligo-X	15.0 ± 0.3 ^a^	12.1 ± 0.4 ^a^	5.9 ± 0.1 ^ab^	7.0 ± 0.1 ^a^

**Table 4 life-12-01757-t004:** Effects of drought stress in the presence or absence of algal extracts on peroxidase, superoxidase dismutase, and polyphenol oxidase activities of wheat at first and second stage. Different letters between columns denote that mean values are significantly different (*p* ≤ 0.05) by Tukey’s test, values are means ± SE.

Treatments	Peroxidase (Unit/µg FW)		Superoxide Dismutase (Unit/µg FW)		Polyphenol Oxidase (Unit/µg FW)	
	First Stage	Second Stage	First Stage	Second Stage	First Stage	Second Stage
Control	76.5 ± 7.5 ^b^	144 ± 12.0 ^b^	36 ± 12.0 ^e^	180 ± 8.4 ^e^	1.2 ± 0.6 ^b^	13.2 ± 0.6 ^b^
Drought Stress	86.5 ± 8.5 ^a^	165.5 ± 8.5 ^a^	60 ± 12.0 ^a^	300 ± 4.2 ^a^	2.4 ± 1.2 ^a^	16.2 ± 1.8 ^a^
*C. elongate*	45 ± 5.2 ^e^	140.5 ± 3.5 ^c^	50 ± 9.3 ^b^	174 ± 13.8 ^e^	2.1 ± 1.5 ^a^	6.6 ± 1.21 ^d^
*S. latifolium*	56.5 ± 2.3 ^d^	106.5 ± 13.5 ^e^	46 ± 9.9 ^c^	240 ± 4.8 ^b^	1.5 ± 3.3 ^ab^	10.6 ± 2.12 ^c^
Canada Power	55.5 ± 3.3 ^d^	138 ± 18.0 ^c^	41 ± 6.0 ^d^	219 ± 6.9 ^d^	1.25 ± 0.3 ^b^	14.3 ± 2.25 ^ab^
Oligo-X	65.5 ± 2.3 ^c^	124.5 ± 7.5 ^d^	39 ± 3.6 ^de^	228 ± 2.4 ^c^	2.1 ± 1.2 ^a^	8.4 ± 1.62 ^d^

**Table 5 life-12-01757-t005:** Effect of foliar spraying of various algal extracts on the phytohormones (IAA, GA3, and ABA) in a wheat plant grown under drought stress. Different letters between columns denote that mean values are significantly different (*p* ≤ 0.05) by Tukey’s test, values are means ± SE.

Treatment	IAA (mg/100 g)	GA3 (mg/100 g)	ABA (µg/100 g)
Control	0.4 ± 0.02 ^b^	9.6 ± 0.03 ^a^	0.5 ± 0.01 ^b^
Drought Stress	0.3 ± 0.05 ^b^	7.5 ± 0.5 ^b^	0.9 ± 0.01 ^a^
*C. elongate*	1.1 ± 0.1 ^a^	9.4 ± 0.4 ^a^	0.4 ± 0.03 ^b^
*S. latifolium*	0.6 ± 0.05 ^b^	8.9 ± 0.3 ^ab^	0.6 ± 0.05 ^ab^
Canada Power	0.5 ± 0.05 ^b^	8.7 ± 0.4 ^ab^	0.3 ± 0.08 ^b^
Oligo-X	0.4 ± 0.01 ^b^	8.8 ± 0.1 ^ab^	0.3 ± 0.09 ^b^

**Table 6 life-12-01757-t006:** Effect of foliar spraying of various algal extract on the morphological traits of wheat yield grown under drought stress. Different letters between columns denote that mean values are significantly different (*p* ≤ 0.05) by Tukey’s test, values are means ± SE.

Treatment	Wt. of the Spike (g)	Spike Length (cm)	No. of Grains/Spike	Wt. Grains (g)/Spike
Control	3.4 ± 0.4 ^a^	11.7 ± 0.4 ^a^	49.9 ± 4.8 ^b^	1.6 ± 0.3 ^a^
Drought Stress	1.6 ± 0.2 ^b^	8.9 ± 0.4 ^b^	38.7 ± 3.3 ^c^	0.9 ± 0.1 ^b^
*C. elongate*	3.2 ± 0.3 ^a^	11.6 ± 0.4 ^a^	54.6 ± 4.2 ^a^	1.8 ± 0.1 ^a^
*S. latifolium*	3.1 ± 0.2 ^a^	11.3 ± 0.5 ^a^	47.7 ± 3.03 ^b^	1.6 ± 0.2 ^a^
Canada Power	2.9 ± 0.2 ^a^	11.1 ± 0.4 ^a^	46.4 ± 2.6 ^b^	1.6 ± 0.1 ^a^
Oligo-X	2.8 ± 0.2 ^a^	11.1 ± 0.6 ^a^	45.7 ± 2.9 ^b^	1.5 ± 0.2 ^a^

## Data Availability

The data presented in this study are available on request from the corresponding author.

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
