# Peer review of "Alleviate the Drought Stress on Triticum aestivum L. Using the Algal Extracts of Sargassum latifolium and Corallina elongate Versus the Commercial Algal Products"

_life, 2022, doi:10.3390/life12111757_

Round 1

Reviewer 1 Report

1.        Language and grammar of the manuscript has ample scope for improvement. Some of the points for example in this regard are:

a.        Line 60: Replace “considers” with “is considered”

b.        Line 70:  Add a comma before “particularly”

c.         Line 72: Replace “for” with “in”

d.        Line 73: Delete “those”

e.        Line 87: Replace “seaweed of” with “seaweeds, viz.

f.         Line 96: Replca “weed” with “weeds”

g.        Line 121: add a space between “H2O” and “before”

h.        Line 122-123: Replace “…and collected the supernatant…” with “…and the collected supernatant…”

i.         Line 128: Replace “…achieved under field conditions…” with “…conducted…”

j.          Line 152: Delete”…to measure the shoot dry weight.”

2.        Section 2.1 (Materials and Methods): Line 112-116: The geographic coordinates of the locations may be given for precision.

3.        Section 2.2 (Materails and Methods: Line 119: Change “oven-dried” to “oven drying”

4.        Section 2.2 (Materails and Methods: Line 120: Change “pulverzied” to “pulverizing”

5.        Line 123-125: Please specify whether the preparation of the commercial algal extract (Canada power and Oligo-X) was in accordance with the instructions from the manufacturer.

6.        Can you please provide a reference to the method of surface sterillization of wheat seeds you employed? Was it standardized by your authorship? If yes, please give the reference.

7.        Employing CRD for field experiments may be justified. Such experiments are laid out in a Radomized Block Design (RBD) with suggested modifications, if any. 

Author Response

Dear Reviewer, many thanks for your valuable comments. We answered all comments point by point as shown in the uploaded author response. 

Reviewer 2 Report

Introduction Section

The introduction is well-conceived and supported with references, but some changes are required. For example,

1.       From lines 67 to 74, these lines do not fit here, and the introduction synthesis is distorted.

2.       Lines 98 to 106 do not present the objectives of the study rather they are explaining the methods. Placing these lines in objectives is wrong. Please consider to re-write.

Materials and methods

1.       Lines 138-142 the treatments are not clear. Please explain whether the control (non-drought) treatment received the algal extracts or not.

2.       Wheat crop normally requires 5-7 irrigations. If irrigations are repeated after every 7 days (as in control) and the crop duration is 167 days. How many total irrigation cycles have been repeated? Also, under drought conditions, the crop receives more than 7-8 irrigations therefore, I believe it is not serious drought stress applied to the crop.

3.       In the abstract, it is mentioned that the crop has been harvested two times (on day 37 and day 67). Is it possible to get two harvests of wheat?? In the methods section, these harvests are replaced with stage I and stage II. Please be uniform.

4.       Was there any fertilizer applied in control and treatments other than the algal extracts?

5.       What do you mean by terminal buds? In my opinion, wheat does not possess terminal buds.

6.       What is the purpose of plant harvest at stages I and II, while normally it is harvested once the crop is matured.

7.       Statistical analysis should be re-written because the order of analytical tests is not correct. The analysis of variance comes first, and LSD comes after ANOVA.

Results and Discussion

1.       In subsection 1 (Lines 226-229), the results are started with a discussion which is merely a definition of drought. The discussion requires justification of obtained results by putting pieces of evidence, reasons, and logical interpretations.

2.       Please do not repeat the treatments, for example, names of commercial products (lines 230-231) and fresh seaweed extracts (lines 231-232) because these names have already been explained in materials and methods and abstract.

3.       The study does not compare different application methods of extracts, but the authors seem to justify the foliar application method (from lines 233 to237). It could make it scientifically robust if the authors justify the effects of extracts rather than application methods.

4.       Figure 1 does not show the LSD (values or letterazation) so significant differences may be identified. From the columns in Fig 1 it is obvious that all the treatments have produced almost equal results except drought treatment. This contrasts with the statement claimed by the authors in line 243 and 245.

5.       “untreated drought-stressed plants…….is this control treatment?? Line 246.

Conclusion

The conclusion is well-written and justifies the objectives of the study.

Author Response

(The authors gave the same response as above.)

Reviewer 3 Report

Nel paragrafo 2.4.1. puoi inserire fasi fenologiche della scala BBCH o della scala Zadoks

Nella tabella 1, 2, 4 e 5 è possibile inserire una o due cifre significative dopo il punto per tutti i valori e la deviazione standard dei trattamenti

Author Response

(The authors gave the same response as above.)

Reviewer 4 Report

The manuscript 'Alleviate the drought stress on Triticum aestivum L. using the algal extracts of Sargassum latifolium and Corallina elongate versus the commercial algal products' shows a low scientific level, errors and shortcomings. Lack of statistical analysis and shallow description of the 'Materials and methods' section. There are a lot of mistakes, e.g. 'draught', 0.9±0.001 means 0.900±0.001 or what?; in a few sections I do not see the values given in the text on the charts, etc.

Manuscript needs a thorough rewritting and refinement of the style of a scientific paper.

Author Response

(The authors gave the same response as above.)
